# Unraveling the Potential of β-D-Glucans in *Poales*: From Characterization to Biosynthesis and Factors Affecting the Content

**DOI:** 10.3390/life13061387

**Published:** 2023-06-14

**Authors:** Michaela Havrlentová, Václav Dvořáček, Lucie Jurkaninová, Veronika Gregusová

**Affiliations:** 1Department of Biotechnology, Faculty of Natural Sciences, University of Ss. Cyril and Methodius, Námestie J. Herdu 2, 917 01 Trnava, Slovakia; veronika.gregusova93@gmail.com; 2National Agricultural and Food Center—Research Institute of Plant Production, Bratislavská cesta 122, 921 68 Piešťany, Slovakia; 3Crop Research Institute, Drnovská 507, 161 06 Prague, Czech Republic; dvoracek@vurv.cz; 4Department of Food Science, Faculty of Agrobiology, Food and Natural Resources, Czech University of Life Sciences, Kamýcká 129, 165 00 Praha, Czech Republic; jurkaninova@af.czu.cz

**Keywords:** β-D-glucans, cereals, biosynthesis, genes, breeding, environment, functions

## Abstract

This review consolidates current knowledge on β-D-glucans in *Poales* and presents current findings and connections that expand our understanding of the characteristics, functions, and applications of this cell wall polysaccharide. By associating information from multiple disciplines, the review offers valuable insights for researchers, practitioners, and consumers interested in harnessing the benefits of β-D-glucans in various fields. The review can serve as a valuable resource for plant biology researchers, cereal breeders, and plant-based food producers, providing insights into the potential of β-D-glucans and opening new avenues for future research and innovation in the field of this bioactive and functional ingredient.

## 1. Basic Characterization and Localization of β-D-Glucans in the Plant

The (1,3-1,4)-β-D-glucans (hereafter referred to as β-D-glucans) are relatively small part components of the cell wall of vegetative tissues of cereals and grasses [1,2]. Mainly cereal grains such as barley (*Hordeum vulgare* L.), oat (*Avena sativa* L.), and rye (*Secale cereale* L.) are rich sources of β-D-glucans, while wheat (*Triticum aestivum* L.), rice (*Oryza sativa* L.), and maize (*Zea mays* L.) dispose of lower concentrations of this polysaccharide [3,4,5,6]. 

Generally, the accumulation of β-D-glucans is observed in the cell wall of endosperm cells of the developing grains and in the surrounding maternal tissues, the aleuronal and subaleuronal layer [7,8,9] (Figure 1). β-D-glucans are not uniformly distributed in the grain, their localization varies among plant species and different plant tissues [8,10]. This polysaccharide has also been found in vegetative organs of the plant, namely in the root, coleoptile, stem, and leaf [11,12,13], and variability in the content of this polysaccharide was found during plant and tissue development [13,14]. 

The β-D-glucans are unsubstituted, unbranched polysaccharides composed of β-D-glucopyranosyl monomers polymerized by both β-(1,3) and β-(1,4) linkages [15,16] and therefore the cereal β-D-glucans are also called “mixed-linkage glucans—MLG”. From a functional point of view, the most important feature of this molecule is the arrangement of β-(1,3) and β-(1,4) linkages along the polysaccharide chain [1,2,17]. The bonds are not arranged in regularly repeating sequences, but they are also not arranged randomly [17,18,19]. (1,4)-β-bonds are usually more often presented in the polysaccharide than (1,3)-β-bonds. (1,3)-β-D-glucosyl residues always occur in linear β-D-glucans chain as individual parts between (1,4)-β-D-oligoglucosyl units, which are mostly found in sequences of two or three [20]. 

The cereal β-D-glucan chain consists of glucopyranosyl monomers linked by a β-(1-4)-glycosidic bond in blocks of three or four monomers, which are called cellotriosyl and cellotetraosyl units. These units are separated by a single β-(1-3) bond, giving the chain a “staircase”-like structure [21,22] (Figure 2). Adjacent (1,3)-β-D-glucosyl residues are not present, at least not in cereal β-D-glucans [1]. Adjacent (1,4)-β-D-oligoglucosyl units located between individual (1,3)-β-D-glucosyl residues can be considered cellodextrin units, which usually consist of two or three adjacent (1,4)-β-D-glucosyl residues [1]. Most β-D-glucans in grasses have longer cellodextrin units, which consist of 5 to 20 adjacent (1,4)-β-D-oligoglucosyl units, which together make up to 10% of the polysaccharide chain [1]. Therefore, β-D-glucans in grasses can be considered (1,3)-β-linked copolymers of cellotriosyl units, cellotetraosyl units, and longer (1,4)-β-D-oligoglucosyl units in which the ratio of cellotriosyl (DP3) to cellotetraosyl (DP4) units (the ratio of β-(1-3) to β-(1-4) units) ranges from 1.5 to 4.5 depending on the source of β-D-glucans [19] with an exception in sorghum endosperm having the ratio 1.15:1 [3]. In barley, the ratio is 2.2 to 2.6:1 [1] or 1.8 to 3.5:1 [23], in wheat the ratio is 3.0 to 4.5:1, in rye it is 1.9 to 3.0:1, and in oats it is 1.5 to 2.3:1 [3,23]. 

The degree of polymerization (DP) of common β-D-glucans in grasses is about 1000 or more [25]. It is a unique feature of the β-D-glucans of each cereal which affects the solubility and viscosity and thus the physicochemical properties and applications of the polysaccharide in solution. For example, oat β-D-glucans with a lower molar ratio (1.5–2.3:1) are more soluble than barley and wheat β-D-glucans with a higher molar ratio (2.6:1 and 3.2:1, respectively) [26]. Differences can be observed in the same genera as genotypic variability [2]. Environmental factors such as the conditions of cultivation also affect the degree of polymerization, whereas oat varieties disposing of higher content of β-D-glucans and cultivated in drier environment show lower degree of polymerization [27]. 

The β-D-glucans found in cereals share the same molecular structure regardless of which source they are isolated from, but certain characteristics are specific to the source of this molecule. These characteristics are, for example, the presence and amount of long cellulose fragments, the ratio between β-(1-4) and β-(1-3) linkages, molecular size, and the ratio of cellotriosyl and cellotetraosyl units [28]. The molecular weight is approximately 31–2700 × 103 g/mol for barley, 65–3100 × 103 g/mol for oat, 21–1100 × 103 g/mol for rye, and 43–758 × 103 g/mol for wheat [23]. For sorghum, it is 36 × 103 g/mol [29]. In the case of DP3:DP4 ratio, a narrow range is, for example, observed in domestic cultivars of *Avena sativa* L. (2.05–2.11) compared to other cultivars of the genus *Avena* (1.81–2.33) [30]. The relative amount of the trisaccharide (DP3) in β-D-glucans decreases from wheat (67–72%) to barley (52–69%) and oats (53–61%), while the relative amount of tetrasaccharide (DP4) has the opposite effect trend, the growth from wheat (21–24%), through barley (25–33%) and oats (34–41%) [23]. Some structural differences between soluble and insoluble β-D-glucans show the DP3:DP4 ratio being higher for insoluble than for soluble β-D-glucans [15,21]. For example, water-soluble β-D-glucans from barley endosperm consist of about 72% of (1,4)-β-glucosyl residues and 28% of (1,3)-β-glucosyl residues [2]. However, comparing the results is difficult because the concept of insoluble β-D-glucans differs from study to study. In any case, this ratio defines the “fingerprint” of the structure of cereal β-D-glucans [26,31]. 

A clear and understandable result of the structural features in β-D-glucans from *Poaceae* is that polysaccharides have (1,3)-β-bonds embedded at irregular intervals along the whole β-D-glucan chain. These bonds cause irregularly distributed molecular kinks in the polysaccharide, which not only prevent the extensive intermolecular arrangement of chains into well-structured microfibrils, but also lead to the formation of polysaccharides that are able to form a gel-like matrix in cell walls and are capable of solubility in water despite its relatively high molecular weight [1,17]. Barley β-D-glucans assume an extended conformation with an axial ratio (length–width) of about 100 in aqueous media [25]. The gel-like structure allows the polysaccharide to provide some degree of structural support to the cell wall, but remains flexible, resilient, and porous enough to allow the transfer of water, nutrients, and other small molecules through the wall during plant growth and development [1]. 

β-D-glucans containing blocks of adjacent (1,4)-β-bonds may tend to aggregate between chains (and thus lower the solubility) through strong hydrogen bonds along the cellulose segments. On the other hand, (1,3)-β-bonds divide the regularity of the (1,4)-β-binding sequence, making the polysaccharide more soluble and flexible. The ability of a cell to change the ratio of β-cellotriosyl and β-cellotetraosyl residues provides a mechanism by which the solubility of a polysaccharide can be fine-tuned and adapted to biological requirements [32]. On the other hand, it is stated that a helix consisting of at least three cellotriosyl residues would represent a stable crystal structure in β-D-glucan molecules; it is therefore possible that a higher content of cellotriosyl fragments could cause some conformational regularity in the chain of β-D-glucans, and thus a higher degree of organization of these polymers (i.e., low solubility) [15]. 

The heterogeneity of the fine structure of β-D-glucans—the ratio of β-(1,4) to β-(1,3) bonds (DP3:DP4) and special distribution of bonds along the chain—obtained by chemical analysis has important implications for the physicochemical properties such as rheological behavior. The most important rheological properties of β-D-glucans include solubility in aqueous solutions and the ability to form a viscous environment [33]. Thus, moderate cellotriosyl:cellotetraosyl ratios (e.g., 1.5 to 2.5:1) would meet functional requirements on a wall such as a porous matrix, while much higher or lower ratios would characterize conformationally more regular, less soluble β-D-glucans, which would have an increased capacity for aggregation with other molecules of β-D-glucans or with cellulose and other cell wall polysaccharides, such as heteroxylans and others [2]. The solubility of β-D-glucans, an important parameter of their functional activities, is also associated with higher content of OH- groups in the structure and so high affinity to water molecules and ability to dissolve in the medium [33].

## 2. Content of β-D-Glucans in Grains of *Poales*

Several studies have been focused on the content of β-D-glucans in cereals such as barley and oat grains as a good natural source of this polysaccharide. Generally, barley varieties contain higher amounts of β-D-glucans compared to oat varieties; however, quality and properties of both β-D-glucans are different (Table 1). Despite this, wheat is not considered to be a good source of β-D-glucans because it has a much lower content, usually <1% on a dry basis. 

Barley is an important cereal grain consumed throughout the world that can be used to develop functional food products rich in β-D-glucans [38,47]. The content of β-D-glucans in barley is on average 3–4% to 8% [38,39], although barley cultivars with the content of 2–11% are observed, and the polysaccharide is in the cell wall distributed throughout the endosperm [8].

Current research of barley β-D-glucans attempts to identify suitable barley genotypes for use in a number of breeding applications such as human nutrition, livestock feed, malting, and brewing [41]. Both varietal variability in the β-D-glucans content and their degradation during germination of barley play a significant role in their application for malting and brewing. Residual malt β-D-glucan from incomplete degradation of endosperm cell walls during the malting process is associated with increased worth viscosity that can slow filtration and reduce brew house efficiency [44]. 

In addition to the genetic resource pools of cultivated barley, wild barley (*Hordeum spontaneum* L.) offers considerable potential as a genetic resource for barley β-D-glucans improvement. A comparative study of the β-D-glucan content between cultivated and wild barley confirmed the higher range and variability of this parameter in wild species. The β-D-glucan contents of the studied wild barley accessions ranged from 3.26% to 7.67% while the content of cultivated barley varieties ranged from 2.68% to 4.74% [41]. 

Another recent large-scale analysis of 117 accessions of wild barley (*Hordeum vulgare subsp. spontaneum* L.) which were selected from ICARDA’s gene bank to represent 21 countries scattered along the natural geographic distribution of the species were carried out by Elouadi et al. [48]. The contents of β-D-glucans ranged from 1.44% to 11.30% in the *Hordeum spontaneum* accessions compared to 36 cultivated barley lines with contents ranging from 1.62% to 7.81%. On the other hand, a similar range (3.6–7.4%) of β-D-glucan contents was already detected by Austrian researchers in 86 hull-less forms of cultivated barely [40]. 

Generally, naked barley has higher starch and β-D-glucan levels than hulled barley. Additionally, mutations at the Lys3 and Lys5 loci can affect β-D-glucan content as well as other health-promoting compounds [49]. While varieties with high β-D-glucans content are preferred in human nutrition, varieties with low concentration are preferred for malting and for broiler fattening. The reliable production of low-β-D-glucan malt based on a suitable barley genotype could replace viscosity control by changing the brewing processes or adding exogenous enzymes. Therefore, novel barley β-D-glucan endohydrolase (β-glucanase) alleles with increased thermostability, e.g., from *Hordeum spontaneum* would be perspective to identify [44]. 

Oat is another major source of β-D-glucans among cereals. The oat usually contains 3 to 5% of this viscous and soluble fiber component [50,51]. Unlike barley grain, the main part of β-D-glucans is located in the thick cell walls in the region of the subaleural outer endosperm [8,10,34]. Therefore, core fractions consisting of subaleural layers are particularly high in this polysaccharide. The content of β-D-glucans in oat grain ranges from 2.3% to 8.5% [4,35] depending on the cultivar and other factors [4,36]. The content of β-D-glucans in diploid oat ranges from 2.85% to 6.77%, in tetraploid it ranges from 3.58% to 5.12%, and in hexaploid oat species the range is 2.88–5.90% [37]. 

A significantly lower variability in β-D-glucan contents was confirmed among four oat species *A. sativa* (3.60%), *A. byzantina* (3.40%), *A. abyssinica* (2.46%), and *A. strigosa* (2.97%) by the recent study of VIR oat collection by Popov et al. [52]. These results indicated that the hexaploid cultivated oat species *A. sativa* and *A. byzantina* showed a generally higher content of β-D-glucans than tetraploid and diploid wild accessions. At the same time, Loskutov and Polonkiy [53] reported a slightly higher β-D-glucan content in naked forms of *A. sativa* compared to hulled oat varieties, which was also confirmed by Havrlentova and Kraic [4]. Thus, it seems that compared to barley species, the variability of β-D-glucan content in oats is not as high and probably not sufficiently mapped. 

In addition, the requirements for the specific β-D-glucan content of oat grain are not as clearly defined as in the case of malting barley processing or poultry fattening. A recent study [54] even reported that 1,3-β-D-glucan can be added to broiler feed to improve the development and integrity of the gut and enhance the immune status of birds without affecting their growth rate. However, barley β-D-glucans do not seem to have this potential and their negative effect on poultry performance is being further studied [55].

## 3. Biosynthesis of β-D-Glucans in *Poales*

The starting point for molecular genetic approaches to the study of cell walls of *Poales* was the sequencing of the rice genome and the subsequent identification of the superfamily of cellulose synthase genes (Cellullose Synthase A, *CesA*), while these genes are responsible for the synthesis of the hexose polysaccharide framework and cellulose itself. Cellulose Synthase-Like genes (*Csl*) were gradually discovered [56]. This family is further divided into eight subclasses: *CslA*, *CslB*, *CslC*, *CslD*, *CslE*, *CslF*, *CslG*, and *CslH*, each containing multiple genes [57]. Later, the *Csl* superfamily was supplemented by the *CslJ* family [58]. In general, *CesA* genes encode cellulose-synthesizing enzymes and *Csl* genes are responsible for the biosynthesis of hemicellulosic polysaccharides [56]. The *CslC* gene group encodes an enzyme that controls the synthesis of the xyloglucan backbone [59], while the *CslF* and *CslH* subfamily genes mediate the synthesis of β-D-glucans in *Poaceae* [60,61], with the *CslF* subfamily being key [60]. The *ClsJ* subfamily is also involved in the biosynthesis of β-D-glucans [1,58], but its phylogenetic significance is still unclear [62]. Not all *Csl* subfamilies are represented in higher plants. The *CslB* and *CslG* subfamilies are only found in dicots and gymnosperms, and *CslF* and *CslH* are only found in monocots [58], with the *CslF* branch diverging from the *CslD* subfamily, which evolved by diversification of *CslD* and *CesA* [63]. Putative functions of the candidate genes involved in the metabolism of β-D-glucans and responsible for co-regulated metabolite network of such polysaccharides in cereals such as starch and β-D-glucans are presented in Figure 3.

The biosynthesis of β-D-glucans, a unique polysaccharide of grass cell walls, is encoded by three groups of *Csl* genes [64], namely *CslF, CslH* and *CslJ* [63], while research has shown that selection pressure on *CslF* subfamily genes is related to the ratio DP3:DP4 in the molecule of polysaccharide [65]. The involvement of the *CslH* subfamily was demonstrated by genetic transformation of *Arabidopsis thaliana*, in which *CslH* genes from rice were inserted and the plant produced β-D-glucans. Similar research was also carried out with the *CslF* subfamily [66]. Likewise, research suggests that the *CslJ* subfamily encodes the synthesis of β-D-glucans in barley, and this subfamily has been observed in barley, wheat, sorghum, and maize, but not in rice [58]. All genes from the *CslF, CslH*, and *CslJ* subfamilies mediate β-D-glucan synthesis in heterologous expression systems [60,61,67], although it is unclear whether each the gene in these groups controls the synthesis of β-D-glucans in vivo [32]. Phylogenetically, the *CslJ* and *CslH* subfamilies existed before the monocot–dicot split and are now present only in monocots [63]. 

In barley tissues, the *CslF6* gene is the most transcribed in terms of β-D-glucan content [68,69,70], while it fulfils the same function in wheat, oats, and rice [68,71]. Knocking out the *CslF6* gene from its function in rice plants caused not only a significant decrease in the content of β-D-glucans in the coleoptile, but also an increased activation of mechanisms related to plant protection against bacterial infections [71]. 

**Figure 3 life-13-01387-f003:**
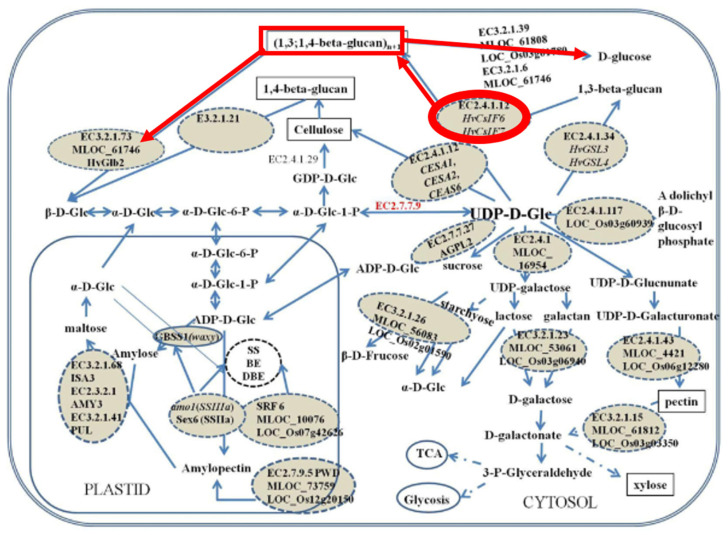
The candidate genes and their encoding protein are dash circled. The cell wall components are marked by black box. The dash arrows indicate there was not a single step. EC3.2.1.73:(1,3; 1,4)-β-glucanase; EC3.2.1.21: β-1,4-glucosidase; EC3.2.1.39: glucan-1,3-β-glucosidase; EC3.2.1.6: glucan-1,3-β-glucosidase; EC2.4.1.12: Cellulose synthase; EC2.4.1.34: glucan synthase; EC2.4.1.117: dilichyl-phosphate-β-glycosyltransferase; EC2.4.1: galactosyl transferase; EC2.7.7.9: UTP–glucose-1-phosphate uridylyltransferase; EC2.7.7.27: ADP-glucose pyrophosphorylase; EC3.2.1.23: β-galactosidase; EC23.2.1.26: β-fructofuranosidase; EC2.4.1.43: glucosyl transferase; EC3.2.1.15: polygalacturonase; EC2.7.9.5: phosphoglucan water dikinase; EC3.2.1.68: Isoamylase; EC3.2.2.1: amylase; EC3.2.1.41: amylopullulanase. The main points of the biosynthetic pathway of β-D-glucans are shown in red [72].

The expression levels of *CslF* subfamily genes intensively studied in different barley tissues and at different developmental stages of the grain showed significant differences in expression between samples. The *HvCslF6* gene is expressed in almost all tissues and during the entire grain development [73], but, e.g., the *HvCslF3* gene is predominantly expressed in the coleoptile and root hairs [73,74], *HvCslF9* is expressed in the coleoptile, roots, and developing seed 8–10 days after pollination, and *HvCslF7* is expressed in the stem and stalk. Based on these results, it can be concluded that the expression of *CslF* genes is regulated by tissue-specific factors. The strict regulation of *CslF* gene expression is particularly visible during seed development, where at least four genes are expressed at very specific stages. These conclusions suggest a multistep process of β-D-glucan biosynthesis, with genes early in the biosynthetic pathway expressed first, followed by other members [73]. 

Most of the research on the genes coding the synthesis of β-D-glucans used barley and rice as plant material. For oats, five genes from the *CslF* family are available in the GenBank database: *AsCslF3, AsCslF4, AsCslF6, AsCslF8*, and *AsCslF9*, with *AsCslF6, AsCslF8*, and *AsCslF9* showing higher similarity to barley, and *AsCslF3* and *AsCslF4* showing less conservatism. Expression analysis of genes involved in the biosynthesis of β-D-glucans in grain, leaves, and panicles of oat revealed a high level of expression of several genes in panicles, while neither the *AsCslF4* nor the *AsCslF9* gene was expressed in leaves. All genes of the *CslF* and *CslH* subfamilies were expressed during grain development. Further results revealed that all genes showed increased expression at later stages of grain development, when β-D-glucans are predicted to accumulate more prominently as grain storage polysaccharides [75]. Comparisons with published results obtained from analysis of barley showed that barley, in contrast to oats, shows increased expression of *CslF6* on the first to fourth [73] or sixth day after pollination [75] with a significant decrease on the eigth day after pollination [73], while the expression of *CslF4* and *CslF8* remains low [69]. 

The allohexaploid (2 n = 6 x = 42; AA, CC, and DD are subgenomes) character of oat with a large complex genome is the cause of more complex and complicated molecular–genetic experiments related to the biosynthesis of β-D-glucans [9]. In the work, Coon [75] focused on the variability of expression of *CslF6* gene homologues in species of the genus *Avena*. The genomic sequence of *CslF6* was approximately 5.2 kb, and due to the large differences between individual homologues in the intron regions, the length of the genes may vary. The A-genome ortholog was 5268 bp, the C-genome was 5162 bp, and the D-genome sequence was 5162 bp in length. The *CslF6* gene contained two introns, the first with a length of 1600 bp and the second with a size of 748 bp. The *CslF6* splice site was determined by comparison of *A. sativa* genomic sequences with *CslF6* coding sequences from *H. vulgare*. The coding sequences were highly conserved among the homologs for the analyzed samples of roots, young plants, and mature embryos at the stages of 1–3, 4–6, and 7–9 days after pollination. *CslF6* expression was significantly lower in mature embryos than at any other developmental stage of the embryo. An exception was the species *A. strigosa*, which had intermediate expression of *CslF6* in mature embryos [75]. 

Regarding the localization of β-D-glucan synthesis, the current dogma for cell wall polysaccharide biosynthesis is that cellulose (and callose) is synthesized at the plasma membrane, whereas matrix-phase polysaccharides are assembled in the Golgi apparatus [7]. The novel model indicates that β-D-glucan does not conform to this paradigm, and in various *Poaceae* species, the CslF6-specific antibody labelling is present in the endoplasmic reticulum, Golgi, secretory vesicles, and the plasmatic membrane and the CslH1 to the same locations apart from the membrane [76]. The updated model of the β-D-glucan synthesis shows that CslF6 is the major synthase and is proposed to act similarly to CesAs, synthesizing glucan chains *de novo* (scenario 1). A less likely but possible alternative is that *CslF6* produces cello-dextrins that are joined together at the plasma membrane by yet an unidentified protein via a single β-(1,3)-glycosidic linkage, creating the β-D-glucan chains that are then recognized by the β-D-glucan-specific antibody (scenario 2) [76] (Figure 4).

## 4. Environmental Factors Influencing the β-D-Glucans Content

Differences in β-D-glucan content among species such as oats, barley, wheat, and sorghum are influenced by genetic and environmental factors [77]. In addition, a mathematical model was developed that is able to pertinently assess the importance of several factors such as genetics, agronomic, and environmental factors to the content of β-D-glucans during cultivation of cereals [36]. Data collected from five locations for two years (2008, 2009) analyzing the content of β-D-glucans in 11 oat genotypes identified locality, genotype, and environmental interaction to be the factors influencing the content of this polysaccharide [37]. As summarized by Redaelli et al. [78], water is one of the most important environmental factors affecting β-D-glucans variability. However, its effect is not always unambiguous. On the one hand, Peterson et al. [79] reported an increase in the content of this cell wall polysaccharide when precipitation was insufficient. Doehlert et al. [80] also mentioned a positive correlation between β-D-glucan content and rainfall in July and August. Furthermore, results can be found in which exposure to heat stress and water deficit before harvest increases the amount of the polysaccharide in grains [47,81]. Another study further confirmed that irrigation during oat growth is responsible for the degradation of β-D-glucan content in the grain [79]. 

Thus, the timing of rainfall and the level of stress are likely to play a role. It can be expected that drought-stressed, e.g., “shriveled” grains show a significantly narrower ratio between starch content and other components including β-D-glucans. Excessive rainfall during the ripening stage can potentially trigger covert sprouting of grains, which could result in a considerable reduction in β-D-glucan content. It is widely recognized in barley and oat malt production [82] that germination leads to a significant breakdown of soluble dietary fiber, particularly β-D-glucans. However, there are no explicit studies demonstrating the effect of pre-harvest germination in spikes on changes in β-D-glucan content in oats or barley.

Nitrogen fertilization has been proved to be a factor influencing the level of biochemical compounds in the grain; it leads to an increase in the content of β-D-glucans [83]. Higher levels of nitrogen in the soil and the use of nitrogen fertilizers greatly increase the total content of β-D-glucans in oat and barley grains [47]. On the other hand, refs. [83,84] also confirm the influence of environment on β-D-glucan variability, but in these results, the effect of genotype dominates over that of environmental factors. This was also confirmed by the work of Dvořáček et al. [85], which compared 11 selected yield and nutritional parameters in five oat varieties. The results showed the significantly highest influence of genotype on the variability of β-D-glucans (over 40%), even in comparison with all other parameters evaluated (e.g., content of crude protein, fat, starch, thousand-grain weight, yield). Furthermore, only year was significant for β-D-glucan variability with an effect of about 21%. On the other hand, the influence of locality and conventional or organic farming management was negligible and not significant (Figure 5). 

It can be further assumed that the major influence of environmental conditions on β-D-glucan level is related to the influence of the processes of synthesis transport and deposition of β-D-glucans during ontogenesis. Tiwari and Cummins [47] reported the highest levels of β-D-glucans after anthesis. During the first 15 days after flowering, 70–90% of the total content of β-D-glucans was accumulated at various nitrogen levels [14]. 

Comparison of the content of β-D-glucans in the panicle of hulled and naked oat samples during plant ontogenesis was also described by Hozlár et al. [13]. The accumulation of β-D-glucans in panicles during oat ontogenesis showed an increasing trend, while in all oat varieties, the content increased from 0.69% and 0.34% to 2.23% and 2.22% in both hulled and naked oat, respectively. β-D-glucan content in developing barley grains was monitored by [86]. The authors confirmed the individual time trend of β-D-glucan accumulation during grain maturation in seven tested barley cultivars. 

It can thus be deduced that the variability in β-D-glucan content is strongly dependent on the genotype but also on the timing of water availability, which influences the rate of synthesis and deposition of the main grain storage substances (polysaccharides and proteins) and thus their relative proportions in the grain.

## 5. Function of β-D-Glucans in the Plant Organism

Interest in cereal β-D-glucans has increased after their acceptance as bioactive and functional ingredients in a healthy diet [87,88] in humans and animals. Nevertheless, β-D-glucans play an important role in the structure and functionality of cereal cell walls [26]. The architectonical function in the cell wall and the storage function in the plant seeds as a source of energy for developing sprouts has been suggested [66]; in addition, the protective role of this cell wall polysaccharide against biotic or abiotic stresses is discussed in the literature [89]. 

Initially, it was thought that β-D-glucans serve to store energy in elongated plant cells and seeds, because the content of this glucose cell wall polysaccharide is increased in young tissues [2,17,19]. This functionality of β-D-glucans is based on the fact that the breakdown of β-D-glucans into glucose is relatively simple and involves only two enzymatic steps that allow rapid mobilization of glucose reserves compared to a longer mobilization process of starch reserves [17]. In addition, the localization of this polysaccharide near starchy endosperm is an advantage for this molecule to serve as a source of energy for developing seedlings. It has also been shown that β-D-glucans can be metabolized as an energy source in vegetative tissues during periods of glucose deficiency. This seems to be confirmed by the fact that barley sprouts, when moved from light to dark, show increased expression of β-glucan endohydrolases and glucolases [26,90], suggesting that the plant mobilizes glucose reserves stored in β-D-glucans to compensate the decrease in photosynthetic activity in the dark [91]. In this case, plant tissues, which are not traditionally associated with photosynthetic activity, use β-D-glucans as a storage vector for glucose other than starch (or instead of the starch polysaccharide). An example is *Brachypodium distachon*, the grains of which contain up to 45% of β-D-glucans and only 6% of starch [92] compared to cereals and most wild grasses, which have 30–70% of starch as the main storage carbohydrate in the grain and generally less than 6% of β-D-glucans [93]. However, growing plants to produce seeds is very difficult.

The architectonical role of β-D-glucans stems from the localization of β-D-glucan microfibrils in the cell wall and from the physicochemical properties of this molecule. In type I cell walls found in dicots and some monocots, the cellulose is encased in a gel-like layer of pectin to provide elasticity and stability to the cell wall. Type II cell walls are found in *Poales* species and contain much less pectin, its function being taken over by β-D-glucans together with arabinoxylans [94], which are highly accumulated during cell elongation, when they can make up to 20% of dry weight of the cell wall [95].The gel-like structure allows the polysaccharide to provide structural support for the cell wall, but at the same time remain flexible, elastic, and porous enough for the transport of water, nutrients and other small molecules across the cell wall during plant tissue development [1]. In young developing plant tissues, it is important that cell walls are porous so that water molecules, nutrients and low-molecular-weight hormones can be freely transported between cells and individual tissues. In the specialized conductive tissues necessary for the long-distance transport of water molecules and nutrients through the plant, the walls, on the contrary, must be impermeable. 

In most plant tissues, cell walls are also important for intercellular adhesion. These different functional requirements for cell walls are maintained by the formation of reinforced gel-like structures consisting of cellulose microfibrils that have high tensile strength and are embedded in a gel matrix phase that consists predominantly of non-cellulosic polysaccharides [26]. This matrix provides the cell with flexibility and a certain mechanical support for maintaining the functional properties of the cell wall. 

In *Gramineae*, the metabolism of β-D-glucans is responsible for plant responses to environmental signals within a moderate, physiological range [96]. For example, in the work of Taketa et al. [69], barley variety producing the standard content of β-D-glucans (3.8%) was characterized by better wintering compared to a mutant with knocked enzyme responsible for producing this polysaccharide. Recent studies also point to the potential of β-D-glucans to be involved in defense mechanisms in the *Poaceae* family against selected forms of environmental stresses [89,97,98,99]. The gel-like layer of β-D-glucans in the cell wall can act as a defensive barrier that protects the cell from fungal invasion, but also provides a potential signaling system that indicates when such an attack is taking place. As fungi attack plant cells and release β-glucanases to digest the protective gel layer surrounding the cell, they slowly dissolve the protective layer of β-D-glucans and expose the cytoplasmic membrane itself. The antioxidant activity of β-D-glucans was also described [100], whereby antioxidant compounds are an accepted factor of the plant’s defense system to cope with biotic aggressors such as fungal pathogens [101,102]. The increase or decrease in the biosynthesis and the associated content of β-D-glucans is conditioned by the expression of plant genes involved in both β-D-glucan synthesis and its degradation. After the exposure of plants to external factors such as pests or pathogens, the β-D-glucan metabolism as well as the entire β-D-glucan turnover are affected and become much more complicated. A study in rice mutants knocking out rice genes involved in the biosynthesis of β-D-glucans showed a phenotype that had a spontaneous response to the lesion, presumably suggesting that fibers of β-D-glucans work as a repressor of the signaling cascade targeting programmed cell death [71]. In addition, reduced deoxynivalenol toxin (DON) content in barley genotypes infected with *Fusarium graminearum* was observed in grains with higher content of β-D-glucans [98] and in oat grains artificially infected with *F. graminearum* and *F. culmorum*, where grains containing higher amounts of β-D-glucans showed lover content of DON and pathogenic DNA [89]. 

The function of β-D-glucans in *Poaceae* as well as the evolutionary preservation of this polysaccharide in monocotyledons are still discussed, as this phenomenon of β-D-glucans has not been precisely explained [1,2,17]. The conformational regularity or irregularity of β-D-glucans defines the properties and thus the physicochemical activities in the matrix in the cell wall. It is this conformational irregularity of β-D-glucans that appears to be a feature limited to *Poaceae* polysaccharides and raises the question whether this irregularity is a key feature that has led to widespread acceptance and preservation of β-D-glucans in the walls of *Poaceae* during evolution [63,67,70]. During the adaptation from aquatic to soil environment, plants had to develop a mechanism for their cells to withstand substantial expansion pressures to be able to prevent the rupture of the cytoplasmic membrane and at the same time contain a structure compatible with the growth of cells and cell tissues; the construction of the cell wall itself of photosynthetically active embryophytes is still crucial [103]. The plant cell wall is designed so that its architecture is strong, but at the same time flexible and able to withstand pressure, tension, and various adverse environmental conditions [104]. There is an assumption that the unique structure of β-D-glucans and the extensive presence of this polysaccharide in species of the genus *Poales* brings enormous evolutionary and adaptive advantages to the plant [26] and is the reason that this group of plants shows extraordinary dynamics of various pressures during evolution [62,70] and inhabits often extreme and inhospitable habitats. 

## 6. Molecular Markers as Breeding Tools for β-D-Glucans Manipulations

In the case of β-D-glucans, the effectiveness of the standard breeding process is related to the identification of suitable genetic resources, knowledge of the responsible genes, the availability of genetic markers, and a good estimate of the heritability of the trait. Efficient phenotyping tools for screening estimation of the β-D-glucan parameters in early generations are also an advantage. 

There are many studies demonstrating the moderate to high heritability of β-D-glucans found in oats and barley. The average heritability level (h2b) of β-D-glucans was estimated to be 0.55 in individual oat plants [105]. An even higher range of (h2b = 0.75–0.84) in the case of β-D-glucans was found in barley [106]. Holthaus et al. [105] also mentioned that groat β-D-glucan content in oat polygenically is controlled primarily by genes with additive effects. Genetic variation for β-D-glucans seemed adequate for effective selection, and genotype–environment interaction was minor. Swanston [107] reported that β-D-glucan content is a quantitative trait with several associated QTLs; one QTL is located on chromosome 7H and is within 5 cM of the *Nud* gene in the case of barley. 

Using current molecular technologies, putative genes responsible for β-D-glucan variation in the grain are continuously studied based on the still limited information on β-D-glucan synthesis. However, gene families involved in the synthesis of these polysaccharides have been identified and include the Cellulose-synthase-like (*Csl*) genes [42], which were described in detail in the previous chapter on β-D-glucan biosynthesis. Members of the *CslF* and *CslH* gene families and Glucan synthase-like (or Callose synthases) are prime candidates for β-D-glucan synthesis. Nevertheless, the *Csl* gene families do not completely explain the variation in β-D-glucan content in cereal grains. There is accumulating evidence that several other genes, including those that hydrolyze β-D-glucans, contribute significantly to their content in mature grains [43]. 

New studies of genome-wide association study (GWAS) associating genetic mutations with measures of β-D-glucan content in cultivated barley have become a powerful tool in identifying candidate genes for selective breeding. Seven putative candidate genes encoding some enzymes in glucose metabolism were found to be associated with β-D-glucan content. One of the putative genes, *HORVU6Hr1G088380*, could be an important gene controlling barley β-D-glucan content [108]. Marcotuli et al. [42] identified seven genomic regions associated with β-D-glucans in a tetraploid wheat collection, located on chromosomes 1A, 2A (two), 2B, 5B, and 7A (two). An analysis of marker trait associations (MTAs) in syntenic regions of several grass species revealed putative candidate genes that might influence β-D-glucan levels in the endosperm, possibly via their participation in carbon partitioning. Walling et al. [43] applied GWAS to the Wild Barley Diversity Collection (*H. spontaneum*) and identified a total of 13 quantitative traits loci (QTL) spread across the seven barley chromosomes that explained most of the variation in β-D-glucan content. Transcriptional dynamics of two barley genotypes differing in grain β-D-glucan content during grain development was investigated by the authors of [109]. Twenty-two differentially expressed genes (DEGs) affecting β-D-glucan accumulation during late developmental stages were selected. Most of these DEGs (encoding alpha-amylase inhibitor, glucan endo-1,3-beta-glucosidase, and sugar transporter) showed different expression patterns in the two genotypes, which might explain the genotypic difference in changes in β-D-glucan content 21–28 days post-anthesis (DPA). 

The hexaploid structure of the oat genome further complicates the detection of β-D-glucan-associated markers. However, Newell et al. [110] associated three DArT markers with β-D-glucan concentration in oats. These markers had sequence homology to rice; one of these DArT sequences, opt.0133, was located on rice chromosome seven and was, by our definition, adjacent to the *CslF* gene family. A recent large GWAS analysis carried out on an oat panel with 413 genotypes was evaluated for β-D-glucan content under subtropical conditions [111]. Seven quantitative trait loci (QTL) associated with β-D-glucan content were identified and located on Mrg02, Mrg06, Mrg11, Mrg12, Mrg19, and Mrg20. The QTL located on Mrg02, Mrg06, and Mrg11 seem to be genomic regions syntenic with barley.

Genome editing methods for influencing β-D-glucan content in barley were used by Garcia-Gimenez et al. [74]. The authors used CRISPR/Cas9 to generate mutations in members of the *Csl* gene superfamily that encode known (*HvCslF6* and *HvCslH1*) and putative (*HvCslF3* and *HvCslF9*) β-D-glucan synthases. Grains from *CslF6-2* (homozygous) mutants almost completely lack β-D-glucans (0.11%), whereas grains from *CslF6-2/+* (heterozygous lines) have intermediate levels of β-D-glucans (1.45%) compared with the wild type control (5.00%). Their data further indicated that only *HvCslF6* from multiple members of the *CslF/H* family showed impact on the abundance of β-D-glucans in mature grain. Genome editing procedures in case of decrease in β-D-glucan content overcome traditional breeding practices and could be a good breeding strategy for the new malting varieties of barley. 

Molecular methods in combination with effective phenotyping procedures for β-D-glucan content (e.g., modified McCleary enzymatic determination or NIRS) are key for the development of new varieties with defined content of the monitored polysaccharide in the breeding process [112,113]. NIRS models developed by Paudel et al. [113] declared high index of determination (h2 ≥ 0.93) and low standard error of cross-validation (SECV ≤ 0.23) for β-D-glucan quantification in ground and whole oat groats as well. The above findings thus hold great promise for obtaining new varieties of oats and barley with declared β-D-glucan content according to the requirements of processors.

## 7. The Role of β-D-Glucans in Brewing Processes

As indicated in the previous sections, barley β-D-glucans play a crucial role in beer production, and monitoring their levels is essential for optimizing the malting process and ensuring high-quality beer. Lower levels of β-D-glucan content in grains and higher levels of β-glucanase in malted barley are associated with better malting performance. If barley has a high starting β-D-glucan content, it can hinder the degradation of cell walls and the diffusion of enzymes during kernel mobilization, leading to disruptions in malt quality parameters [114]. 

Higher levels of β-D-glucans can contribute towards issues such as haze formation, viscous wort, and reduced wort filtration during the brewing processes. Malt extract is a complex quantitative trait that is controlled by multiple genes and its concentrations can be variable in different cultivars. It is also considered as a mega-trait which is the product of interactions between many sub-traits [115,116].

Breeders face challenges in genetically manipulating and selecting malting quality traits such as ME and β-D-glucan content due to their complex inheritance patterns. Numerous studies have identified over 250 QTLs associated with malting quality, including malt extract and β-D-glucan content. These QTLs have been found and mapped on different chromosomes of barley. Among these QTLs, QTL2 on chromosome 4H stands out as a major contributor to barley malting quality, accounting for 29% and 38% of the variation in key parameters such as β-D-glucan content and malt extract, respectively [117].

In a separate study, the telomeric region of chromosome 4H, which contains the complex associated with malting quality, underwent fine mapping. This investigation revealed the presence of 15 potential QTLs related to key malting quality parameters such as the content of β-D-glucans, ME, alpha-amylase, and diastatic power (DP) [118]. Numerous endeavors have been undertaken to develop molecular markers for the purpose of selecting barley varieties with improved malting traits, including malt extract and β-D-glucan content. For example, a study involved genotyping of approximately 1524 single-nucleotide polymorphisms (SNPs) to identify several genes associated with six malting traits, including β-D-glucan content and malt extract [119]. In recent research, QTL2, a locus that exhibits a substantial proportion of variation in malt extract and content of β-D-glucans, was selected. This locus contains a key gene, *HvTLP8*, which appears to play a role in the interaction with β-D-glucans in a redox-dependent manner [120]. 

The levels of β-D-glucans in the wort are influenced by the malting process itself, specifically through the action of the β-glucanase enzyme. The enzyme is predominantly synthesized in the aleurone and scutellum of germinated grains. As the malting process progresses, there is a significant decrease in β-D-glucan content due to the increased activity of this enzyme. This reduction in the concentration of the polysaccharide relies on the initial lower levels of the polysaccharide in the grain and higher quantities of the β-glucanase enzyme in the resulting malt. The activity of β-glucanase during malting leads to hydrolysis of cell walls, converting them into soluble β-dextrin with a low molecular weight. It is important to note that β-glucanase is sensitive to temperature; it is rapidly inactivated at temperatures above 50 °C during the extraction phase. However, the breakdown of β-D-glucans from intact cell walls continues, resulting in the accumulation of soluble β-D-glucan molecules in the extract. Therefore, the final characteristics of the produced malts are also influenced by factors such as temperature, humidity, and the duration of germination [121].

## 8. Conclusions

The review emphasizes the unique characteristics of β-D-glucans, highlighting their composition of β-D-glucopyranosyl monomers connected by β-(1,3) and β-(1,4) bonds, resulting in a distinctive “staircase” structure. It further underscores that β-D-glucans are predominantly located in cell walls in selected cereal grains such as barley (*Hordeum vulgare* L.) and oats (*Avena* ssp. L.), playing a significant role in the structural integrity, functionality, and defense mechanisms of cell walls. Moreover, the review delves into the genetic and environmental factors influencing β-D-glucan content in cereals, with particular emphasis on genotype, water availability, and nitrogen fertilization. It highlights the contemporary relevance of molecular markers and genome-wide association studies in facilitating precise breeding and genome editing approaches for manipulating β-D-glucan levels, offering promising avenues for enhancing crop traits. 

## Figures and Tables

**Figure 1 life-13-01387-f001:**
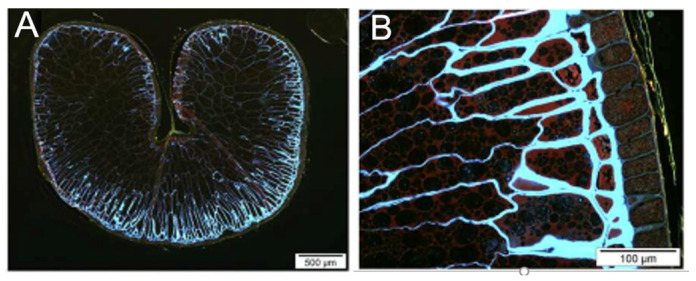
Localization of β-D-glucans in grains of *Avena sativa* variety SW Betania. Vertical (**A**) and longitudinal (**B**) seed sections show the presence of the polysaccharide in blue by calcofluor staining and visualized in a fluorescence microscope, adapted from [10].

**Figure 2 life-13-01387-f002:**
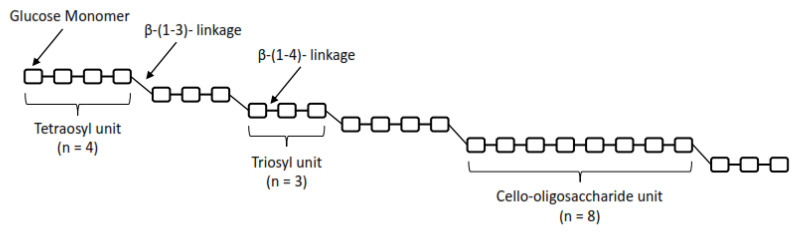
Basic structure of cereal β-D-glucans with β-(1-3)- and β-(1-4)- linkages and the “staircase”-like structure, adapted from [24].

**Figure 4 life-13-01387-f004:**
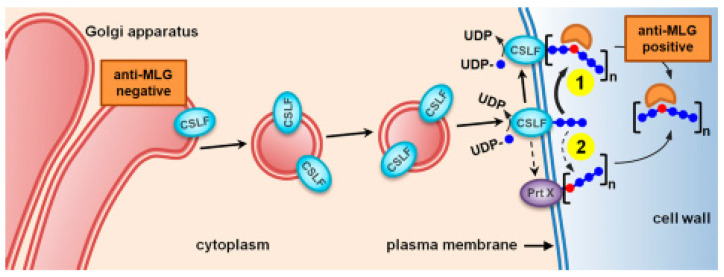
Novel model of the possible synthesis of β-D-glucans in cereals. CSLF6 is the major MLG synthase that is proposed to act similarly to CesAs, synthesizing *de novo* the MLG chains that are recognized by MLG-specific antibody at the cell surface (scenario 1). A less likely but possible alternative is that *CSLF6* produces cello-dextrins (shown here in blue) that are joined together at the plasma membrane by yet an unidentified protein (Prt X) via a single b-(1,3)-glycosidic linkage (shown in red), creating the MLG chains that are then recognized by the MLG-specific antibody (scenario 2) [76].

**Figure 5 life-13-01387-f005:**
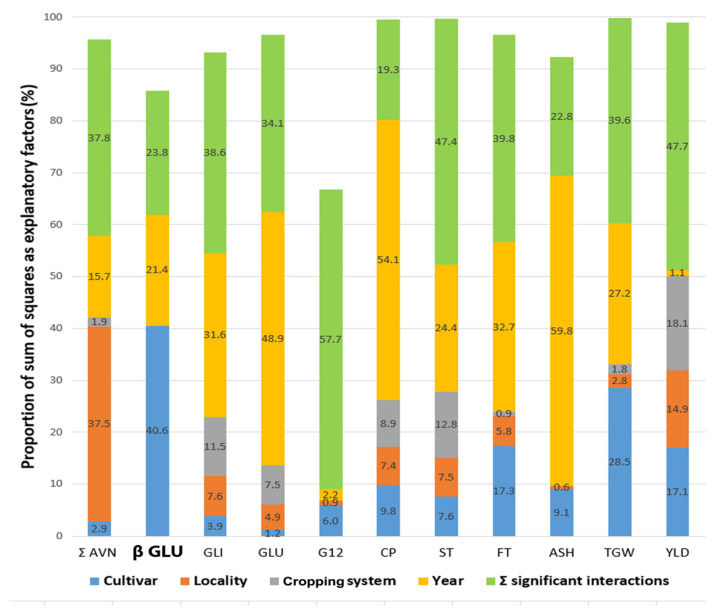
The effect of the main significant factors and interactions influencing the variability of oat grain β-D-glucans in comparison with other significant oat grain parameters. Calculations were performed based on the factorial proportion of the sum of squares according to the analysis of variance (ANOVA) model for selected grain parameters of five oat varieties grown at two locations and under different cropping systems (organic vs. conventional) for 3 years. Total content of avenanthramides (Σ AVN), β-D-glucans (β GLU), gliadins (GLI) and glutelin (GLU) protein fractions, immunoreactive avenin peptides (G12), crude protein (CP), starch (ST), fat (FT), ash (ASH), thousand-grain weight (TGW), and yield (YLD) was evaluated [85].

**Table 1 life-13-01387-t001:** Differences in the structure and properties of the β-D-glucans among selected cereal sources.

Plant Species	Oat (*Avena sativa* L.)	Barley (*Hordeum vulgare* L.)	Rye (*Secale cereale* L.)
β-D-Glucans			
Localization in the *grain*	Outer layers of the endosperm and starchy endosperm, variable among varieties [8,10,34]	Uniformly throughout the endosperm [7]	-
Level of β-D-glucans (% dwb)	2.3–8.5 [4,35,36,37]	2.68–8 [38,39,40,41]	1.2–1.6 [6]
Fiber structure		Extended conformation with an axial ratio of about 100 in aqueous media [25]	
DP3:DP4	1.5–2.3:1 [3,23]	1.8–3.5:1 [1,23]	1.9–30:1 [23,33]
Relative amount of DP3 in the molecule (%)	53–61 [22]	52–69 [22]	-
Solubility	Higher [26]	Lower [26]	-
Molecular size (g/mol)	65–3100 × 10^3^ [22]	31–2700 × 10^3^ [22]	21–1100 × 10^3^ [22]
Heritability	0.55 [42]	0.75–0.84 [43]	
Breeding trends	Increase in β-D-glucan content for human nutrition	Decrease in β-D-glucan content in malting barleys [1] and barleys for poultry fattening [44]. Increase in β-D-glucan content for human nutrition [45,46]	Increase in β-D-glucan content for human nutrition
Application	Human nutrition, functional ingredient, functional foods, livestock feed	Human nutrition, livestock feed, malting, brewing [38,41,47]	Human nutrition, functional foods, functional ingredient

## Data Availability

The data presented in this study are available on request from the corresponding author.

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
