# Peer review of "Unraveling the Potential of β-D-Glucans in Poales: From Characterization to Biosynthesis and Factors Affecting the Content"

_life, 2023, doi:10.3390/life13061387_

Round 1
Reviewer 1 Report
Dear Authors,
please, see the attached file.
Best regards.
Reviewer.

Author Response
Dear reviewer, thank you very much for reviewing the manuscript and thank you for your useful remarks. Your final decision was "not to publish", however we as authors made corrections in the whole text to improve the quality and also we responded to your decision. We really and strongy believe that you will accept our revision and answers.
On behalf of the authors team Michaela Havrlentova

Reviewer 2 Report
The review manuscript is interesting and update important information about β-Glucans.
Pg. 11, Line 512 – The reference “(EFSA Journal 2010;8(12):1885)” should be numbered in text like the other ones.
I would like to suggest to authors to include a small topic about β-Glucans problems in some food and beverages industrial products. For example, it is known that β-Glucans in barley, malt, wort and beer have long been implicated in malting and brewing process problems.
Author Response
Dear reviewer, thank you very much for your time, energy and useful remarks to improve the quality of the manuscript. We accepted all your suggestions.
On behalf of the authors team Michaela Havrlentová

Reviewer 3 Report
Review
Cereal β-D-glucans: From biosynthesis and breeding possibilities to food processing applications.
The authors reviewed cereal β-glucans from biosynthesis and breeding possibilities to food processing applications. This review covers the wide-range of aspects of β-glucans. It should be useful, but it might be difficult to read for readers of other fields.
Some parts of this manuscript are inconsistent. For example:
L46:"not arranged randomly"; L109: "The random distribution".
L32: The highest concentrations of β-D-glucans are found in the outer layers of the grain.; L142: The polysaccharide is mainly distributed in the grain endosperm (75%) and aleurone layer (25%).
Some sections, for example, in section 7.2., the authors list lots of references but no interpretations or discussions of these results.
Fig.1: It is difficult to understand. It should be focused on β-glucans.
Fig.2: This model does not show which reactions are more important for β-glucan synthesis than others.
L545: Do “cultivars” mean barley cultivars?
Table 1: It simply lists reference. It needs to be summarized for each use.
The term of “β-D-glucan” should be used throughout this manuscript. β-glucan was also used on L194 and L308.
I also suggest the authors include more figures, e.g., the structures of β-glucans and food products for better understanding for general readers. It is also helpful to include a table comparing differences among barley, oat and rye in terms of β-glucan.
Author Response
Dear reviewer, thank you very much for your time, energy, and useful remarks to improve the quality of the manuscript. We accepted all your suggestions.
On behalf of the authors team Michaela Havrlentová

Round 2
Reviewer 1 Report
Dear Authors,
two little remarks.
Figure 2: n=3 and n=4 mixed up.
7. The roll of β-D-glucans in brewing processes. roll or role?
Best regards.
Author Response
Dear Reviewer,
on behalf of the author's team, thank you very much for preparing the review, which will contribute to the higher quality of the manuscript.
Suggestion 1: Figure 2: n=3 and n=4 mixed up.
Answer 1: Thank you for this comment. The picture / structure was taken over directly from the cited work. We really did not see this mistake. Now, the structure is corrected.
Suggestion 2: 7. The roll of β-D-glucans in brewing processes. roll or role?
Answer 2: Thank you very much. Of course, the role is correct. We corrected the title of this chapter.
Once again, thank you for your time, energy, and suggestions to improve the quality of the manuscript.
With regards.
Reviewer 3 Report
Review
Cereal β-glucans: From biosynthesis and breeding possibilities to food processing applications.
Table 1: For breeding trends for barley, increase in β-D-glucan content for human nutrition should be included. The following references should be cited:
, & (2019). Breeding naked barley for food, feed, and malt. In I. Goldman (Ed.), Plant breeding reviews (1st ed., pp. 95– 119). Wiley. https://doi.org/10.1002/9781119616801.ch4
Tonooka, T. Yanagisawa, T., Emiko Aoki, et al. (2022) Breeding of a new six-rowed waxy barley cultivar “Kihadamochi” exhibiting high levels of yield and β-glucan content. Breeding Research. 24:146-152.
Fig.1: There are no explanation why three oats are shown in this figure. I suggest to compare oats, barley and rye to show variations of their localization among plant species.
Fig.3: The title should be revised. EC2.1.4.12 in legend should be EC2.4.1.12. This figure is not well-organized. Enzymes and encoding proteins, which don't contributed to β-D-glucan synthesis can be omitted. EC2.4.1.12 and HvCsIF6 should be highlighted as a major MLG synthase.
Fig.5: The legend does not explain AVN 2p, AVN 2f, AVN 2c, AVN 2pd, AVN 3p, AVN 3f, AVN 2fd, AVN 5f, and GU. These data are neither explained in the manuscript. The authors should reconsider this figure. ΣAVN should be enough in this manuscript.
L338: "hidden pre-harvest sprouting" should be reconsidered with scientific evidences.
L594-595: "β-D-glucans are predominantly located in the outer layers of cereal grains" should be revised.
Author Response
Dear reviewer, thank you very much for your time, energy, and useful comments to improve the quality of the submitted manuscript.
Here are the answers to your comments and suggestions:
Review
Cereal β-glucans: From biosynthesis and breeding possibilities to food processing applications.
Suggestion1: Table 1: For breeding trends for barley, increase in β-D-glucan content for human nutrition should be included. The following references should be cited:
Meints, B., & Hayes, P. M. (2019). Breeding naked barley for food, feed, and malt. In I. Goldman (Ed.), Plant breeding reviews (1st ed., pp. 95– 119). Wiley. https://doi.org/10.1002/9781119616801.ch4
Tonooka, T. Yanagisawa, T., Emiko Aoki, et al. (2022) Breeding of a new six-rowed waxy barley cultivar “Kihadamochi” exhibiting high levels of yield and β-glucan content. Breeding Research. 24:146-152.
Answer 1: Thank you for this comment. Both citations we found, read carefully, and citations we added to the text, to the table 1.
Suggestion 2: Fig.1: There are no explanation why three oats are shown in this figure. I suggest comparing oats, barley, and rye to show variations of their localization among plant species.
Answer 2: Thank you for the suggestion. It would certainly be useful for the reader of the article to see a comparison of the localization of glucans in oats, barley, and rye in one picture. Unfortunately, we could not find a suitable image that would demonstrate the given phenomenon. To make the informative value of the image - comparison of localization of the glucan in different cereals sources - high, it would be necessary to use the same staining and microscopy technique. We could not find such images. Therefore, we allow ourselves to retain only the localization of glucans in oat grain as one example of cereals sources of glucan. The title of the image was modified, as well as the description of the image in the text of the manuscript. Thank you in advance for your understanding.
Suggestion 3: Fig.3: The title should be revised. EC2.1.4.12 in legend should be EC2.4.1.12. This figure is not well-organized. Enzymes and encoding proteins, which don't contributed to β-D-glucan synthesis can be omitted. EC2.4.1.12 and HvCsIF6 should be highlighted as a major MLG synthase.
Answer 3: Thank you for your responsible reading of the manuscript and its pictured. The labelling EC2.1.4.12 was corrected and also highlighted. We allowed ourselves to keep the diagram, as it shows all the connections in the synthesis of important substances in the cell. So that the whole scheme is complete and whole for the reader. I hope, you will accept this decision. Thank you very much in advance.
Suggestion 4: Fig.5: The legend does not explain AVN 2p, AVN 2f, AVN 2c, AVN 2pd, AVN 3p, AVN 3f, AVN 2fd, AVN 5f, and GU. These data are neither explained in the manuscript. The authors should reconsider this figure. ΣAVN should be enough in this manuscript.
Answer 4: The picture was corrected, parts of AVN were removed. The legend of this picture was rewritten. Thank you for this important comment.
Suggestion 5: L338: "hidden pre-harvest sprouting" should be reconsidered with scientific evidences.
Answer 5: This sentence was corrected, and one new citation was added.
Suggestion 6: L594-595: "β-D-glucans are predominantly located in the outer layers of cereal grains" should be revised.
Answer 6: This sentence in the conclusion part was corrected. Thank you very much for this important remark.
Dear reviewer, once again, on behalf of the authors team I would like to thank you very much for your useful remarks and suggestions to improve the quality of the manuscript.
With regards.
